# Compression Reconstruction and Fault Diagnosis of Diesel Engine Vibration Signal Based on Optimizing Block Sparse Bayesian Learning

**DOI:** 10.3390/s22103884

**Published:** 2022-05-20

**Authors:** Huajun Bai, Liang Wen, Yunfei Ma, Xisheng Jia

**Affiliations:** 1Shijiazhuang Campus, Army Engineering University of PLA, Shijiazhuang 050003, China; bai_huajun@sina.com (H.B.); lwenmark@163.com (L.W.); fcz1992@sina.com (Y.M.); 2Hebei Key Laboratory of Condition Monitoring and Assessment of Mechanical Equipment, Shijiazhuang 050003, China

**Keywords:** diesel engine, data compression, vibration signal, K-SVD, fault diagnosis

## Abstract

It is critical to deploy wireless data transmission technologies remotely, in real-time, to monitor the health state of diesel engines dynamically. The usual approach to data compression is to collect data first, then compress it; however, we cannot ensure the correctness and efficiency of the data. Based on sparse Bayesian optimization block learning, this research provides a method for compression reconstruction and fault diagnostics of diesel engine vibration data. This method’s essential contribution is combining compressive sensing technology with fault diagnosis. To achieve a better diagnosis effect, we can effectively improve the wireless transmission efficiency of the vibration signal. First, the dictionary is dynamically updated by learning the dictionary using singular value decomposition to produce the ideal sparse form. Second, a block sparse Bayesian learning boundary optimization approach is utilized to recover structured non-sparse signals rapidly. A detailed assessment index of the data compression effect is created. Finally, the experimental findings reveal that the approach provided in this study outperforms standard compression methods in terms of compression efficiency and accuracy and its ability to produce the desired fault diagnostic effect, proving the usefulness of the proposed method.

## 1. Introduction

Diesel engines have been widely used in energy, construction machinery, and military equipment. Vibration signals are transmitted dynamically and synchronously in real-time, Playing a pivotal role in real-time online monitoring of diesel engine health [1,2,3]. It can effectively reduce the incidence of equipment failure, downtime, and management costs. Compared with traditional wired data transmission, edge computing wireless data transmission methods can significantly improve the real-time, flexibility, and ease of use of data, according to the Nyquist sampling theorem [4]. To realize the collection of high-frequency vibration signals of the equipment, we will inevitably generate a large amount of data. However, big data is constrained by network bandwidth during wireless transmission. Whether it can support the problems of massive data, high concurrency, low latency, and low power consumption is yet to be determined.

Recently, it has become a research hotspot that researchers focus on. For example, Antonopoulos et al. [5] embedded compression algorithms into hardware systems to improve the work efficiency of transmitting large amounts of data wirelessly. Ma et al. [6] used a distributed video codec scheme to enhance the processing power of a single node for traditional data compression. Yi et al. [7] proposed an adaptive data compression and transmission range extension scheme to improve the data collection rate of sink nodes. Hameed et al. [8] used lossless compression technology and Huffman coding encryption technology to provide effective means for remote monitoring security and compressibility of electrocardiography (ECG) data. Therefore, before the data are wirelessly transmitted, real-time synchronous sampling and compression of the original vibration data is the best solution to solve the above problems.

Compressive sensing (CS) is a new technical theory that has emerged in recent years [9]. Due to its outstanding performance in data compression and reconstruction, it has been widely used in the field of image and sound. Use the observation matrix to map the original vibration signal from the high-dimensional space to the low-dimensional space. Then, the original signal is recovered with a high probability from fewer observations through an optimization algorithm. Currently, commonly used compression and re-construction algorithms include greedy algorithm [10], convex optimization algorithm [11], Bayesian learning [12], etc. For example, Liu et al. [13] used a low-pass filtering method to optimize the electrographic signal and used basis pursuit (BP) algorithm to compress and reconstruct the electrocardiogram signal. Cheng et al. [14] used an improved orthogonal matching pursuit (OMP) algorithm to improve seismic data’s reconstruction speed and compression effect. Sajjad et al. [15] used a genetic algorithm to optimize the sparse signal and the regularized orthogonal matching pursuit (ROMP) algorithm to reconstruct the image signal. Generally, reciprocating mechanical vibration signals have sparse, non-sparse, and unique structural features. The traditional compression and reconstruction algorithm is used to recover sparse signals with high accuracy and versatility in the above research. However, this type of algorithm only considers its sparsity and is not necessarily suitable for reconstructing reciprocating mechanical vibration signals. Improving the recovery accuracy of structured non-sparse signals becomes crucial.

In the existing Bayesian algorithm, the block sparse Bayesian learning bound optimization (BSBL-BO) algorithm [16] has the potential to solve the problem of structured non-sparse signal reconstruction. The algorithm effectively uses the intra-block correlation of vibration signals to restore structured non-sparse signals. Compared with other traditional compression and reconstruction algorithms, the BSBL-BO algorithm has the advantages of high signal recovery accuracy and good compression effect and has been widely used in electrocardiograms and radar. For example, Mahrous et al. [17] proposed a space-time sparse Bayesian learning method. By optimizing the BSBL-BO algorithm, the compression and reconstruction of multi-channel electro-encephalogram (EEG) signals are realized. Li et al. [18] used an enhanced narrow-band interference separation algorithm for radar to achieve compression and reconstruction of radar signals through the BSBL framework, proving the feasibility of the BSBL-BO algorithm for data compression. However, this algorithm has not been studied much in reciprocating mechanical vibration signals in previous studies. This paper carries out related research based on the BSBL-BO algorithm to fill the gap.

An essential prerequisite for CS is the sparsity in the original vibration signal. Sparsity plays a crucial role in the accuracy of the reconstruction of recovered data. Therefore, an efficient data dictionary is needed to improve the signal’s sparsity. Classical dictionaries include discrete cosine transform (DCT) [19], discrete Fourier transform (DFT) [20], and wavelet packet transform (discrete wavelet transform, DWT) [21] are fixed dictionaries. The ideal sparse representation can only be obtained when the atomic features in this dictionary type are the same as the original vibration information. There is also a dictionary, commonly used K-singular value decomposition (K-SVD) [22] and optimal directions (method of optimal directions, MOD) [23]. The dictionary is dynamically updated through training to obtain the optimal sparse representation. Compared with the fixed dictionary, it has the advantage of solid adaptive ability. For example, Li et al. [24] used the K-SVD algorithm to update the dictionary to improve the sparsity of image signals. Yang et al. [25] used the K-SVD algorithm to enhance the sparse representation of medical images to obtain better compression and reconstruction accuracy.

Diesel engines often have various failures in their daily work. Among them, the loss of the diesel engine refers to the phenomenon of increased valve clearance, severe deformation of a valve seat ring, burning oil, and severe wear of piston rings during operation. As a result, the diesel engine cannot work normally, and there is a more significant safety hazard. To reduce the occurrence rate of diesel engine failures and improve stability and safety, researchers have carried out a great deal of research work and achieved fruitful research results. Gu et al. [26] applied the multivariate empirical mode decomposition to the fault diagnosis of diesel engine misfire and achieved good fault classification results by using the SVM classifier. Chen et al.’s [27] harmony search optimizer is used to set hyper-parameters of the variational stacked autoencoder. This method has been well applied in the fault detection of diesel engines. Wang et al. [28] proposed the plan of particle swarm optimization probabilistic neural network (probabilistic neural network, PNN) and support vector machine. Effective diagnosis of common engine failures is achieved. In recent years, the application of compressed sensing theory to fault diagnosis has gradually attracted the attention of researchers, and some research results have been completed. Zhang et al. [29] trained several over-complete dictionaries with a dictionary learning method. Thereby, redundant dictionaries corresponding to different fault categories are obtained. The matching tracking algorithm is used to determine. The error of the reconstructed signal under various dictionaries is compared to realize the diagnosis of the fault category. Tang et al. [30] first obtained the compressed acquisition signal. Then, given the specified sparsity, the matching pursuit algorithm is used to directly obtain the first few fault characteristic frequencies with enormous energy. To realize the identification and diagnosis of fault signals, Du et al. [31] used a dictionary constructed from Fourier transform matrices. The fault features are directly extracted in the compressed measurement domain to realize fault diagnosis of vibration signals.

Although compression technology has been widely used, there are still the following problems or deficiencies:In the process of wireless transmission, due to the limitation of network bandwidth and low power consumption, massive vibration signals bring considerable challenges to data storage and wireless network transmission;The problem of the reconstruction accuracy of the structured non-sparse signal of the reciprocating mechanical vibration signal cannot be satisfied by the traditional data compression technology;Aiming at the compression and reconstruction effects of reciprocating mechanical vibration signals, there is a lack of an effective, comprehensive evaluation index for data compression effects;There is a lack of relevant research on compressive sensing technology and fault diagnosis methods and their application in fault diagnosis of reciprocating machinery.

Using the BSBL-BO algorithm can effectively solve the problem of structured non-sparse signal reconstruction. At the same time, the sparsity of the signal can also be enhanced by the adaptive dynamic updating of the K-SVD dictionary. Combining the two methods can efficiently and accurately recover structured non-sparse signals. Therefore, this paper proposes a compression and reconstruction method based on the BSBL-BO algorithm and the K-SVD dictionary. In addition, this article also establishes an evaluation index for the effect of data compression. First, divide the original signal into blocks. Use the K-SVD dictionary to obtain optimal sparse decomposition to train the actual movement to improve the re-construction performance of the restored signal. Second, use the BSBL-BO algorithm to restore structured non-sparse signals. Compared with other reconstruction algorithms, it has the advantages of high accuracy and a good data compression effect. Finally, the proposed BSBL-KSVD algorithm is verified through a diesel engine valve clearance experiment and fault classification. The experimental results prove that the BSBL-KSVD algorithm proposed in this paper is practical and feasible, providing a reference basis for wireless data transmission of reciprocating mechanical vibration signals.

The main contributions of this paper are summarized as follows:Using the BSBL-KSVD algorithm and exploiting the intra-block correlation of the vibration signal, we can recover the structured non-sparse signal efficiently. Compared with other traditional compression and reconstruction algorithms. We can effectively improve the reconstruction accuracy and compression effect;A comprehensive evaluation index of compression effect suitable for reciprocating mechanical vibration signal is constructed, and it has a good engineering application prospect;We apply compressed sensing technology to fault diagnosis. The wireless trans-mission efficiency of the vibration signal can be effectively improved to achieve a better diagnosis effect and has a better reference value.

The second section of this article describes the diesel engine compression reconstruction method model based on BSBL-KSVD; the third part is the comprehensive evaluation index of vibration data compression effect; the fourth part verifies the effectiveness of the compression reconstruction method through preset failure experiments. Finally, this research is summarized.

## 2. Model of Diesel Engine Compression Reconstruction Method Based on BSBL-KSVD

### 2.1. Compressed Sensing

In traditional data acquisition and transmission, the Nyquist sampling theorem is used. Usually, the sampling frequency is set to more than twice the highest frequency in the signal under test. Due to the high sampling frequency, a large amount of data is generated. This brings considerable challenges to the wireless data transmission, storage, and remote real-time dynamic monitoring of the operational status of the diesel engine. The emergence of CS theory breaks through the limitation of the traditional vibration signal sampling theorem. Combining the acquisition of vibration signals with the compression process, a small number of signals contains most of the valuable data. Assuming the original signal $x\in R^{N\times 1}$ and observation matrix $\Phi \in R^{M\times N}$ (M≪N), then the signal *x* is linearly projected on the matrix $y\in R^{M\times 1}$ as a compressed signal. Then, the compressed observation of the original signal $x\in R^{N\times 1}$ can be obtained [32]:(1)y=Φx+v

Among them, *v* represents the unknown noise vector. The CS algorithm uses the compressed data *y* and the measurement matrix Φ to restore the original vibration signal *x*.

### 2.2. Block Sparse Bayesian Learning Reconstruction Algorithm

We were using the block structure characteristics of sparse signals. Based on the block sparse Bayesian learning framework, data compression can be realized. In actual engineering applications, the signal *x* has a block structure feature, as shown in the following equation [16]:(2)$$X={\left[\underset{X_{1}^{T}}{\underbrace{x_{1},\cdots ,x_{d_{1}}}},\cdots ,\underset{X_{g}^{T}}{\underbrace{x_{d}{}_{{}_{g-1}+1},\cdots ,x_{d_{g}}}}\right]}^{{}^{T}}$$

The model combined by Equations (1) and (2) is called a block sparse data compression model. We use the characteristics of intra-block correlation to improve the ability of compressed data recovery. Therefore, based on the model in the BSBL framework, it is assumed that the independent *x_i_* between each block satisfies a multivariate Gaussian distribution [16]:(3)$$p(x_{i};\gamma_{i},B_{i})∼N(0,\gamma_{i}B_{i}),i=1,\cdots ,g$$

Among them, $\gamma_{i}$ and $B_{i}$ both represent unknown parameter variables. $\gamma_{i}$ represents a non-negative parameter variable that controls the block sparsity of the original signal *x*. $B_{i}$ represents a positive definite matrix used to obtain the related structure between elements in each block. Assuming that the noise vector obeys the Gaussian prior distribution p(v;λ)∼N(0,λI), use Bayesian principle to obtain the posterior probability of *x*, as shown in the following equation [16]:(4)$$p(x|y;\lambda ,{\left\{\gamma_{i},B_{i}\right\}}_{i=1}^{g})∼N(\mu_{x},\sum {}_{x})$$

Among them, $\mu_{x}={\sum_{0}\Phi^{T}(\lambda I+\Phi \sum_{0}\Phi^{T})}^{-1}y$, $\sum {}_{x}={\left(\sum {}_{0}^{-1}+\frac{1}{\lambda }\Phi^{T}\Phi \right)}^{-1}$.

When the parameters λ and ${\left\{\gamma_{i},B_{i}\right\}}_{i=1}^{g}$ are solved, then the maximum posterior estimate of *x* can be obtained as $\hat{x}$. Next, use the second type of maximum likelihood estimation method to obtain this parameter, as shown in the following equation [16]:(5)$$\begin{array}{c} L(\Theta )≜-2\log\int p(y|x;\lambda )p(x;\{\gamma_{i},B_{i}\}i)dx\\ \;\;\;\;\;\;\;\;\;\;\;\;\;\;\;\;\;\;\;\;\;\;\;\;\;\;\;\;=\log|\lambda I+\Phi \sum_{0}\Phi^{T}|+y^{T}(\lambda I+\Phi \sum_{0}\Phi^{T}{)}^{-1}y \end{array}$$
where Θ represents the parameters λ, ${\left\{\gamma_{i},B_{i}\right\}}_{i=1}^{g}$.

### 2.3. K-SVD Adaptive Over-Complete Dictionary

The traditional fixed dictionary has a particular sparse representation when the signal is sparsely decomposed. Since the sparse representation of the limited dictionary is unknown, its suitability and flexibility are not strong enough. To further improve the sparsity, we need to use an adaptive dictionary learning method for optimization. Therefore, the K-SVD learning dictionary is used as the spare base to obtain a better sparse representation. The dictionary atom is dynamically updated through training until an adaptive over-complete dictionary is obtained. To ensure that the atomic scale in the dictionary is closer to the atomic scale in the original signal, the training process of dictionary D is expressed as [33]:(6)$$\min_{D}{\left\|\left.Y-DA\right\|\right.}_{F}^{2}\;\mathrm{s.t.}\;{\left\|\left.a_{i}\right\|\right.}_{0}\leq T$$

In the above equation, Y represents the given training dictionary matrix, A represents a sparse matrix, and *T* represents the sparsity of the sparse representation vector to be solved.

Initialization D belongs to a super-complete dictionary, and there is a certain degree of redundancy. Suppose that when we update the *j*-th column atom in dictionary D, we also let *E_i_* be the calculation error after removing the *i*-th atom; *d_j_* represents the *j*-th column of dictionary D, and *a^i^* represents the *i*-th row of sparse matrix A. Then, the objective function is as follows [33]:(7)$${\left\|\left.Y-DA\right\|\right.}_{F}^{2}={\left\|Y-\sum_{j=1}^{K}d_{j}a^{i}\right\|}_{F}^{2}={\left\|(Y-\sum_{j\neq 1}d_{j}a^{j})-d_{i}a^{i}\right\|}_{F}^{2}={\left\|E_{i}-d_{i}a^{i}\right\|}_{F}^{2}$$

When directly decomposing *E_i_*, the elements in the obtained *a^i^* may not be sparse. Therefore, only the non-zero elements in *a^i^* need to be updated, defined as the following equation:(8)$$w_{i}=\left\{\left.k\left|a^{i}(k)\neq 0\right.\right\}\right.$$
represents the index collection of the index of the non-zero element in ai. The SVD decomposition method is used to update the atomic vector gradually, and the sparse representation coefficient matrix A in the dictionary D. Next, we generate a new dictionary through multiple iterative updates.

### 2.4. Basic Flow of BSBL-KSVD Algorithm

The algorithm flow of compression and reconstruction of diesel engine vibration signal based on BSBL-KSVD is shown in Figure 1. The algorithm mainly includes dictionary training, data compression, and signal reconstruction.

The specific implementation steps are as follows:

**Step 1.** The signal is divided into blocks. Customize the collected original vibration signal *x* into *i* blocks and the size of the elements in each block;

**Step 2.** Dictionary training: Initialize the dictionary parameters, set the number of training samples, use the K-SVD algorithm to train the examples, and obtain an optimized dictionary Ψ;

**Step 3.** Data compression: The vibration signal of reciprocating machinery is more complicated than that of rotating machinery. To further improve the sparsity of the signal, the optimized dictionary Ψ can map the signal to the sparse transformation, and the original signal *x* = Ψ*θ* can obtain the sparse transformation signal *θ*. The sensing matrix *A* = ΦΨ (that is, observation matrix × sparse matrix) compresses the sparse signal data and obtains the data compressed signal observation value *y* = *Aθ*;

**Step 4. Signal transmission:** The block-compressed signals *y*_1_,…,*y_i_* are successively transmitted through the data network;

**Step 5. Signal reconstruction:** After receiving the compressed signal block, using the BSBL-KSVD reconstruction algorithm proposed in this article through the sensor matrix *A*_1_, …, *A_i_* and compressed signal *y*_1_, …, *y_i_* to reconstruct, we obtain the restored sparse signal *θ*_1_, …, *θ_i_*. At the same time, we perform inverse sparse transformation to obtain reconstructed signal blocks *x*_1_, …, *x_i_* and connect the reconstructed signal blocks one by one and finally form a complete reconstructed signal *x′*.

The pseudo code of the algorithm (Algorithm 1) is as follows:
 **Algorithm 1** BSBL-KSVD algorithm pseudo code  1. Input: *x* = [ *x*_1_, *x*_2_, …, *x**_i_*], blkLen, N, M;  2. Initialize dictionary parameters: *param. L* = 5, *param. K* = 70, *param. numIteration* = 20, *param. Initialization Method = ‘Data Elements’*; *group Start Loc = 1:blkLen:N*;  3. K-SVD dictionary training: [Ψ, output] = KSVD(*x_i_*, param); the core is to use Equations (6)–(8) to generate a new dictionary Ψ through multiple iterative updates;  4. Sparse transformation: *x_i_ =* Ψ*θ*;  5. Sensor matrix: *A_i_* = ΦΨ;  6. Using the combination of Equations (1) and (2), the observation value of the data compression signal is obtained *y_i_* = *A_i_θ*;  7. Signal transmission: The block-compressed signals *y*_1_, …, *y_i_* are successively transmitted through the data network;   8. **For** *i* = 1: size(*x_i_*,2)/N (signal reconstruction);  9. *θ_i_* = BSBL_BO (*A_i_*, *y_i_*, *groupStartLoc*, 0, *‘prune_gamma’*, −1, *‘max_iters’*, 20); the core is to use Formula (3)–(5) to solve the reconstructed signal *θ_i_*;  10. Perform inverse sparse transformation to obtain reconstructed signal blocks: *x*_1_, …, *x_i_*;  11. Connect the reconstructed signal blocks one by one to finally form a complete reconstructed signal: *x**′* = *x*_1_ + *x*_2,_ + *…* + *x_i_*;  12. **End**;  13. Output: *x′*.

## 3. Comprehensive Evaluation Index of Vibration Data Compression Effect

Reciprocating machinery vibration signal components are complex when compared to rotating machinery vibration signal components, noise pollution is severe, and a considerable amount of redundant data is created. There are numerous techniques in extant research to solve the data compression challenge. However, innumerable metrics are necessary to evaluate the data compression effect and performance benefits thoroughly. Although data compression technologies are widely utilized in the voice and image sectors, no standardized complete evaluation approach exists. As a result, while researching the vibration data compression method used in reciprocating equipment, it is vital to define a standard for evaluating the data compression effect. The following thorough assessment index of the data compression effect is produced by combining the structural properties of reciprocating equipment vibration data.

### 3.1. Data Compression Rate Evaluation Index

Data compression rate refers to the ratio of compressed data to the original data. It is a straightforward, intuitive, and easy-to-understand key indicator. Use CR (compressing ratio, CR) to represent the data compression ratio, and the range is set to (0, 1); then, the compression ratio is defined as follows [34]:(9)$$\mathrm{CR}=\frac{N-M}{N}$$

N represents the original signal in the above equation, and M represents the compressed signal. The larger the CR value, the higher the data compression rate. When the data compression rate is higher, it does not mean that the data compression and reconstruction effect is better. It needs to be combined with the standard mean square error index for comprehensive evaluation.

### 3.2. Standard Mean Square Error Evaluation Index

The data compression rate is used to evaluate the ability of data compression. It shows that the loss rate of the original signal in the data compression process is very high. The accuracy of the reconstructed original signal is closely related to the compression rate. When the data compression rate is more significant, we cannot accurately restore the original signal reconstructed from the compressed signal. Therefore, based on data compression, MSE (mean square error, MSE) is used to represent the standard mean square error index, which the following equation can calculate [35]:(10)$$\mathrm{MSE}=\frac{\left\|Z^{\prime }-{\left.Z\right\|}_{2}\right.}{\left\|{\left.Z\right\|}_{2}\right.}$$

Z represents the original signal in the equation above, and Z′ represents the reconstructed signal. The smaller the MSE value, the higher the accuracy of the data compression reconstructed signal. When the data compression rate is more significant, the MSE value is smaller, indicating better data compression and reconstruction effect.

### 3.3. Peak Signal-to-Noise Ratio Evaluation Index

The peak signal-to-noise ratio refers to the ratio of the original signal to the data com-pressed and reconstructed signal. In data compression, the loss of data information is reduced, and the quality of retaining the original data is improved as much as possible. PSNR (peak signal-to-noise ratio, PSNR) is used to express the peak signal-to-noise ratio [35], which the following equation can calculate:(11)$$\mathrm{PSNR}=101g(z_{max}^{2}/(\frac{1}{N}\sum_{j=1}^{N}{\left(z_{j}-z_{j}^{\prime }\right)}^{2}))$$

z represents the original signal in the equation above, z′ represents the reconstructed signal, and z*_max_* represents the maximum component. The greater the PSNR value, the higher the accuracy of the data compression and reconstruction signal, the closer it is to the original signal. It shows that the data compression and reconstruction effect is better.

### 3.4. Pearson Correlation Coefficient Evaluation Index

In evaluating the effect of data compression and reconstruction of the signal and using the two indicators of MSE and PSNR, usually, the Pearson correlation coefficient can also be used to evaluate the degree of correlation between the reconstructed signal and the original signal. Use r to represent the Pearson correlation coefficient, and the range is set to (−1,1), which can be calculated by the following equation [36]:(12)$$r_{z,z^{\prime }}=\frac{N\sum ZZ^{\prime }-\sum Z\sum Z^{\prime }}{\sqrt{N\sum z^{2}-{\left(\sum z^{\prime }\right)}^{2}}\sqrt{N\sum {\left(z^{\prime }\right)}^{2}-{\left(\sum z^{\prime }\right)}^{2}}}$$

Z represents the original signal in the equation above, and Z′ represents the reconstructed signal. When the value of r is closer to 1, the similarity between the compressed and reconstructed signal and the original signal is higher and, conversely, the lower the similarity to the actual movement.

### 3.5. Comprehensive Evaluation Index in Time Domain

In fault prediction and health management, extracting characteristic parameters from vibration signals is crucial. Provide input conditions for further relevant analysis. For compressed data, the compression reconstruction algorithm should be able to recover from the compressed reconstructed signal similar to the original signal. Furthermore, in theory, it is identical to the actual feature parameters. Commonly used time-domain characteristic parameters mainly include mean value, root mean square value, variance and peak value, and other 12 indicators [37]. Under the same compression ratio, the feature parameters extracted from the reconstructed signal from compressed data are closer to the feature parameters extracted from the original signal, indicating that the less loss in the data compression process, the better the data restoration effect.

To better reflect the compression effect of the reconstructed signal in the time domain signal, the time domain characteristic index *TT_i_* is defined, which can be calculated by the following equation:(13)$$TT_{i}=\left|\left.\frac{T_{i}-{\tilde{T}}_{i}}{T_{i}}\right|\right.,\;i\in \left\{\left.1,2,3,\cdots ,12\right\}\right.$$

In the equation, *T_i_* represents the time-domain feature value of the original signal, and ${\tilde{T}}_{i}$ represents the time-domain feature value of the reconstructed signal. The smaller the *TT_i_* value, the closer the time-domain characteristic index of the reconstructed signal and the original signal and the more accurate the data compression effect and restoration effect.

Similarly, in order to better evaluate the data compression effect of different compression algorithms, the comprehensive evaluation index *KPI_t_* of time domain characteristics is defined, which can be calculated by the following equation:(14)$$KPI_{t}=\sum_{i=1}^{12}\omega_{i}\cdot TT_{i}$$

In the equation, $\omega_{i}$ represents the weight coefficient, which satisfies $\omega_{i}$ > 0, and $\sum_{i=1}^{12}\omega_{i}$ = 1. If there is no special case, the value is set to $\omega_{i}$ = 1/12, *i* = 1,2,3,…,12. The smaller the *KPI_t_* value is, the closer the reconstructed signal data recovery is to the time domain index of the original signal, and the more accurate the corresponding data compression effect is.

## 4. Experimental Data Verification

### 4.1. Experiment Preparation

Figure 2 is the in-line six-cylinder diesel engine test bench used in the research. The test bench comprises three parts: diesel engine condition monitoring panel, diesel engine, and vibration signal data acquisition system. The diesel engine status monitoring panel can control the ignition, acceleration, and flameout of the diesel engine. The instrument reflects the engine speed, water temperature, voltage, and remaining oil. Preset 6 intake valve clearance state modes under different working conditions include one normal status and five other fault states. The detailed parameters of the dataset are shown in Table 1. To obtain valid data samples, four vibration sensors are arranged on the cylinder head of the diesel engine, as shown in Figure 2b. Among them, the sampling frequency of data acquisition is set to 20 kHz, and the duration of each acquisition is set to 10 s. Each failure mode collects ten sets of data samples, and each data group contains 200,000 points (20 kHZ sampling for 10 s), as shown in Figure 3.

### 4.2. Comparison of BSBL-BO Algorithm with Other Compression and Reconstruction Algorithms

#### 4.2.1. Evaluation Index of Reconstructed Signal MSE under the Same Compression Ratio

Compare and analyze BSBL-BO algorithm with block sparse Bayesian learning-expectation-maximization (BSBL-EM), compressive sampling matched pursuit (CoSaMP), BP, OMP, and ROMP algorithm. Use the Valve_1200_7mm dataset to verify and analyze the reconstruction algorithm, as shown in Figure 4. To ensure the reconstruction performance of the algorithm, the data compression rate is uniformly set to 0.5, and the sparse dictionary matrix uniformly uses the K-SVD generation method. From the analysis results in Figure 4, it can be seen that the smaller the MSE index, the higher the reconstruction accuracy, indicating that under the same parameter setting conditions, the proposed BSBL-BO reconstruction algorithm has more advantages than its reconstruction algorithm. The recovered reconstructed signal is closer to the original signal, proving that the data compression and reconstruction effect is better.

#### 4.2.2. Evaluation Index of Reconstructed Signal MSE under Different Compression Ratios

A comprehensive analysis of the algorithm’s influence on different compression ratio changes is carried out. Six different datasets are used to verify the compression and reconstruction algorithm. Among them, each dataset sets 13 kinds of compression ratios. Each compression rate is performed 100 times of MSE calculation. Find the corresponding variance σ and average μ, and use the 95% confidence interval (μ − 2σ, μ + 2σ) method to express, as shown in Figure 5. It can be seen from the analysis result of Figure 5, when CR < 0.6, the MSE index of the BSBL-BO reconstruction algorithm proposed in this paper is smaller than other reconstruction algorithms. Know the accuracy, superiority, and effectiveness of the proposed method. When CR > 0.6, all reconstruction algorithms have a more considerable MSE value as the compression ratio increases. It means that the data lose essential information during the compression process, resulting in a significant reduction in the reconstruction accuracy. The ROMP algorithm has the most considerable MSE value and the lowest reconstruction accuracy. As the compression ratio increases, the reconstruction accuracy also decreases. Conversely, the lower the compression ratio, the higher the reconstruction accuracy. Therefore, after being verified by six different datasets, under the premise of ensuring a specific data compression rate and sure re-construction accuracy, when the CR = 0.5, it is confirmed that the method proposed in this paper is the best for data compression of vibration signals.

#### 4.2.3. Peak Signal-to-Noise Ratio Evaluation and Pearson Correlation Coefficient under Different Compression Ratios

As shown in Figure 6a, the method proposed in this paper has significant advantages compared with other methods. When the compression ratio increases, the PSNR value decreases, indicating that more data information is lost during data compression. The reconstructed signal is different from the original signal and has a low peak signal-to-noise ratio. Combining them with the MSE metric is recommended when evaluating data compression results. The larger the PSNR index, the smaller the MSE index and the better the data compression effect. As shown in Figure 6b, The BSBL-KSVD method also outperforms other ways and the Pearson correlation coefficient increases as the compression ratio decreases. The results show that much of the original signal’s information is preserved in the data when compressed. Therefore, the reconstructed signal has a high similarity with the original signal. When evaluating the effect of data compression, it is recommended to combine the MSE indicator. The smaller the MSE index, the higher the Pearson correlation coefficient, and the better the data compression effect. From a comprehensive analysis, when CR = 0.5, it is proven that the method proposed in this paper has the best compression effect and is more suitable for data compression of vibration signals.

#### 4.2.4. Comprehensive Evaluation Index of Reconstructed Signal in Time Domain under Different Compression Ratios

Next, to better evaluate the pros and cons of the recovered reconstructed signal, Using the same compression and reconstruction algorithm and data in Section 4.2.1 and combined with the time domain comprehensive evaluation index *KPI_t_* for comparative analysis, the *KPI_t_* index weights are all set to 1/12, and the analysis results are shown in Figure 7. It can be seen from Figure 7 that the smaller the *KPI_t_* index is, it means that the restored reconstructed signal retains most of the original signal. The time-domain characteristics of the reconstructed signal are closer to the frequency domain characteristics of the original signal, which proves that the proposed method has the best data compression effect. In a comprehensive analysis, the corresponding *KPI_t_* index is more minor when the compression rate is lower, indicating that the data compression effect is better. Therefore, it is proven that when the compression ratio CR = 0.5, the compression effect of the BSBL-KSVD algorithm proposed in this paper is optimal, which is more suitable for data compression.

### 4.3. K-SVD Dictionary and Other Dictionary Effect Verification Comparison

In data compression, the sparse representation of the signal is critical since the sparse representation of the static dictionary has relatively low complexity. Assuming that the signal feature information is consistent with the atomic data in the dictionary, a more accurate and effective sparse representation can be obtained. Commonly used classic fixed dictionaries to obtain the sparse dictionary matrix include DFT, DWT, DCT, etc.

Therefore, the K-SVD dictionary is compared and analyzed with the DCT, DFT, and DWT dictionaries. Use the Valve_1200_7mm dataset to verify and scrutinize the reconstruction algorithm. First, the 200,000 sampling points of the original signal only select the first 64,000 sampling points for block compression. The length of each signal block is set to 80 sampling points, which are divided into 800 blocks. A Gaussian random matrix uniformly generates the observation matrix. Secondly, for the K-SVD dictionary, the number of atoms is set to 50, the number of iterations is set to 20, and 300 blocks of signals are trained each time. The remaining 500 pieces of signs are used to verify the validity of the dictionary. Finally, the single variable principle is adopted, and the BSBL-BO compression and reconstruction algorithm is uniformly adopted. It is applied to different sparse dictionaries and verified from the MSE evaluation index, peak signal-to-noise ratio evaluation index, and Pearson correlation coefficient evaluation index.

#### 4.3.1. Evaluation Index of Reconstructed Signal MSE under Different Compression Ratios

As shown in Figure 8a, the compression effect of vibration data based on the K-SVD dictionary is better than that of other dictionaries. The blue lines represents the original signal, and the red lines represents the reconstructed signal in Figure 8b. Observing Figure 8b, we can find that when CR = 0.5, the waveform of the reconstructed signal based on the K-SVD dictionary is closer to the original signal than in other dictionaries. When CR > 0.7, the greater the MSE index, and the data reconstruction effect is worse. Therefore, it is proven that the proposed method is more suitable for data compression when CR = 0.5.

#### 4.3.2. Peak Signal-to-Noise Ratio of Reconstructed Signal under Different Compression Ratios

In Figure 9b, the blue lines represents the original signal, and the red lines represents the reconstructed signal. As can be seen from Figure 9, The data compression effect based on the K-SVD dictionary is better than other dictionaries. When CR > 0.7, the PSNR indicator becomes smaller as the compression rate increases. It shows that a great deal of data information is lost in data compression. The recovered reconstructed signal is quite different from the original signal, and the peak signal-to-noise ratio will naturally become smaller. It needs to be evaluated in combination with MSE indicators. When the MSE index of the reconstructed signal is smaller, and the PSNR index is more extensive, it proves that the performance of the proposed method is better. Therefore, when CR = 0.5, it is more suitable for data compression.

#### 4.3.3. Pearson Correlation Coefficient of Reconstructed Signal under Different Compression Ratios

In Figure 10b, the blue lines represents the original signal, and the red lines represents the reconstructed signal. As can be seen from Figure 10, The data compression effect based on the K-SVD dictionary is also better than other dictionaries. When CR < 0.6, as the compression rate gradually decreases, the more significant the Pearson correlation coefficient, and the data retains a large amount of original signal information during the compression process. The similarity between the restored reconstructed signal and the original signal becomes higher. Therefore, it needs to be used in conjunction with the MSE indicator. When the MSE indicator is more minor, and the Pearson correlation coefficient is more significant, it is proven that the compression effect of this method is the best. When CR = 0.5, it is more suitable for data compression.

## 5. Application of Compressed and Reconstructed Signal in Fault Diagnosis

To further verify the effectiveness of the BSBL-KSVD compression reconstruction method proposed in this paper in fault diagnosis, two forms of fault classification accuracy are adopted: naive Bayes classifier (NBC) and support vector machines (SVM). A comprehensive evaluation is performed to check the quality of the compressed and reconstructed signal. The higher the classification accuracy, the closer the reconstructed signal is to the original signal. The fault test dataset in Table 1 is used for fault diagnosis, and ten sets of samples are taken for each fault state. There are 200,000 sampling points in each group, with 5500 sampling points as a group, divided into 360 groups of samples and six failure states, a total of 6 × 360 = 2160 samples. Each sensor’s fault state is extracted from the time and frequency domains, including 22 characteristic parameters such as mean values, root mean square values, variance, and peak values [37,38]. Each sensor forms a 22 × 2160 feature matrix.

### 5.1. Comparative Analysis of Fault Classification under Different Compression Ratios

Therefore, select sensor 1–4# data to form a feature matrix of 88 × 2160. After dimensionality reduction by the stacked sparse autoencoder (SSAE) method, SSAE input nodes are set to 88, and the hidden layer parameters are 50 and 22, respectively. The sparsity ratio is set to 0.1, the weight adjustment coefficient is set to 0.000002, and the sparsity penalty weight is set to 0.0002. A new 22 × 2160 feature matrix is obtained, divided into 1800 training samples and 360 test samples. Using the built-in classification learning tool of Matlab 2020. Among them, 1800 training samples adopt the K-fold cross-validation method and take K = 10. Input to the classifier method: NBC and SVM are trained, and the optimal training model is obtained. Then, input 360 test samples into the trained model for fault identification. Obtain the results of fault diagnosis accuracy, as shown in Table 2. The confusion matrix of the fault diagnosis results is shown in Figure 11. It can be seen from the effects that the higher the compression rate CR value, the lower the fault classification accuracy rate. SVM has an accuracy rate of 96.39% for the original signal fault diagnosis, while NBC has an accuracy rate of 90.83%. When CR = 0.25, the classification accuracy of SVM reaches 95.56%, while that of NBC is 89.72%. It is very close to the classification result of the original signal. We obtain the same conclusion as in Section 4.2: The BSBL-KSVD compression reconstruction method is suitable for high data compression.

As shown in Figure 11, whether it is the original signal or different compressed signals, the fault recognition rates for fault 1, fault 2, and fault 3 are relatively low. Among them, the classification result of defect one increases with the increase of compression rate, while the accuracy rate gradually decreases. Therefore, the BSBL-KSVD compression reconstruction method proposed in this paper hopes to find the optimal balance between the fault diagnosis accuracy and the wireless network transmission. It shows that this kind of fault signal contains fewer fault features, which increases the difficulty of fault classification. It can be recognized if the fault diagnosis accuracy rate is more than 90%. Then, when CR = 0.5, the compressed vibration signal during wireless transmission will significantly reduce the constraint of network bandwidth and improve the transmission efficiency.

### 5.2. Comparative Analysis of Fault Diagnosis Results of Different Compression and Reconstruction Methods

We compare and analyze BSBL-KSVD with other compression and reconstruction algorithms and use the experimental data in Section 4.1 to verify the method’s effectiveness. First, use five compression and reconstruction algorithms to process the original data in Table 1 with three compression ratios (i.e., CR = 0.25, CR = 0.50, CR = 0.75). Then, using the feature extraction method in Section 5, different fault feature matrices of 88 × 2160 are extracted from the other reconstructed signals of the four sensors. The SSAE method is also used for dimensionality reduction where the SSAE parameter settings are the same as in Section 5.1. Finally, for the three compression ratios under each compression and reconstruction method. We can obtain a new 22 × 2160 feature matrix after dimension reduction, respectively, and divide it into 1800 training samples and 360 testing samples. In addition, use the built-in SVM classification tool of Matlab 2020 for fault diagnosis. The relevant parameter settings are the same as those in Section 5.1, and the final diagnosis results under different compression ratios are shown in Table 3, Table 4 and Table 5.

From the diagnostic results in Table 3, it can be seen that when CR = 0.25, the diagnostic results of the BSBL-KSVD method are better than other compression and reconstruction methods. The diagnostic results of BP, ROMP, and OMP algorithms are less than 90%, indicating that the reconstruction accuracy of these three types of strategies is not high. Some critical data information is lost during data compression.

From the analysis of the diagnostic results in Table 4, when CR = 0.5, only the diagnostic results of the BSBL-KSVD method are > 90%, The diagnostic results of the other four compression methods were lower than 90%, and the lowest diagnostic result of the OMP method was only 77.33%. It shows that with the increase of compression ratio, the diagnosis result gradually decreases.

From the results in Table 5, when CR = 0.75, the diagnostic results of BP, ROMP, and OMP methods are all below 80%, while the BSBL-KSVD method can reach more than 90%. Compared with other compression methods, the method proposed in this paper has good robustness and superiority.

To sum up, the diagnosis results of the BSBL-KSVD method are better than other compression and reconstruction methods under different compression ratios. In the case of weighing various pros and cons, it is assumed that the diagnostic result is >90% and has a high data compression rate. This is a good reference for applying data compression to mechanical fault diagnosis.

## 6. Conclusions

This paper proposes a method of compression and reconstruction of diesel engine vibration signal based on BSBL-KSVD, which is practical and feasible, and compared with other methods, there are advantages. To effectively verify the pros and cons of the BSBL-KSVD algorithm proposed in this study regarding data compression effects, use the CR indicator, MSE indicator, PSNR indicator, r indicator, and KPIs indicator for verification and, finally, compressed and reconstructed signals for fault diagnosis case analysis. The experimental results show that the compression effect of the BSBL-KSVD algorithm is optimal when the compression rate CR = 0.5. The recovered reconstructed signal is closer to the original signal, and good classification accuracy is obtained, which has a good engineering application prospect.

Although the proposed method has achieved good results, we can still improve it in the following aspects: First, this research did not focus on using the reconstructed signal to perform signal repair and noise reduction preprocessing in the follow-up. We will conduct detailed research using methods such as double sparse dictionary learning; second, it did not consider integrating the algorithm with the data acquisition hardware. In the subsequent investigation, embedding the algorithm into FPGA improves front-end data acquisition, transmission performance, and efficiency.

## Figures and Tables

**Figure 1 sensors-22-03884-f001:**
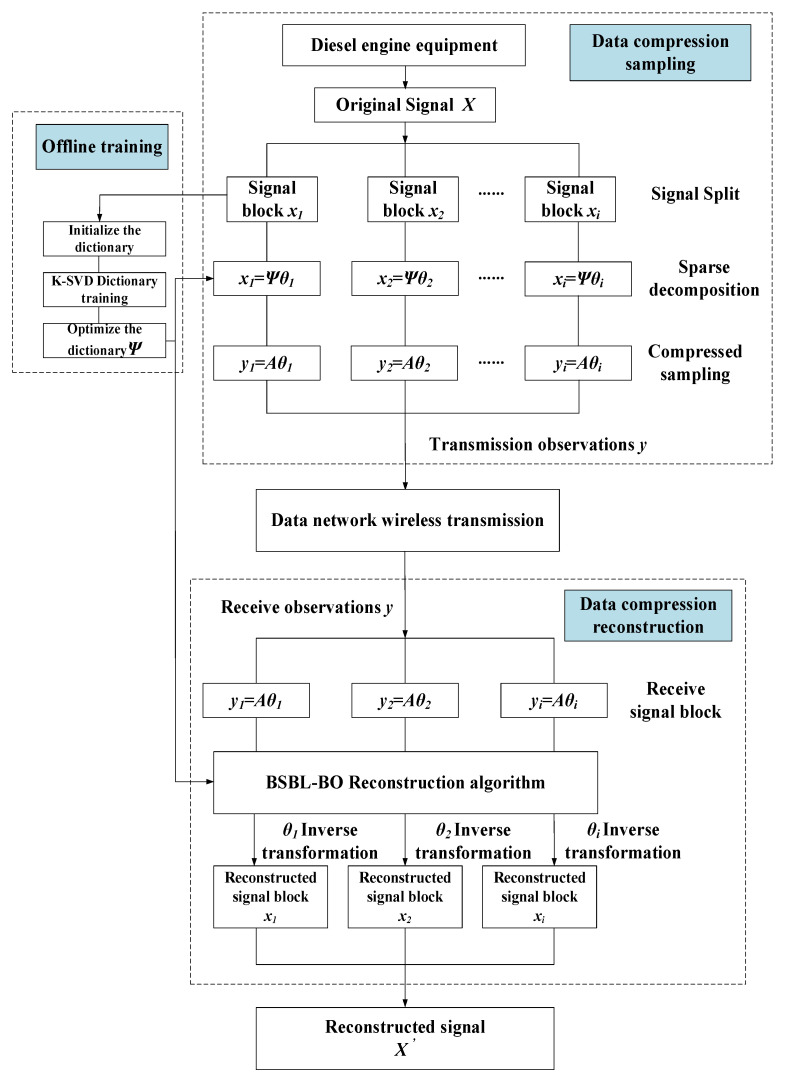
BSBL-KSVD compression reconstruction algorithm flow.

**Figure 2 sensors-22-03884-f002:**
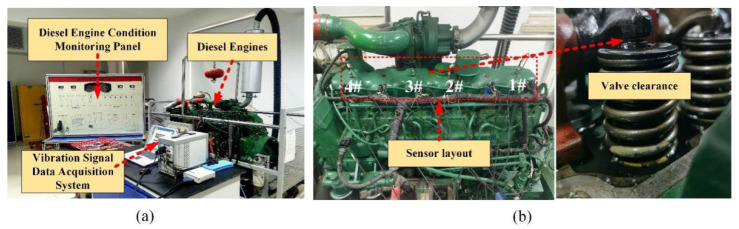
Diesel engine preset failure experiment environment: (**a**) test bench; (**b**) intake valve clearance failure.

**Figure 3 sensors-22-03884-f003:**
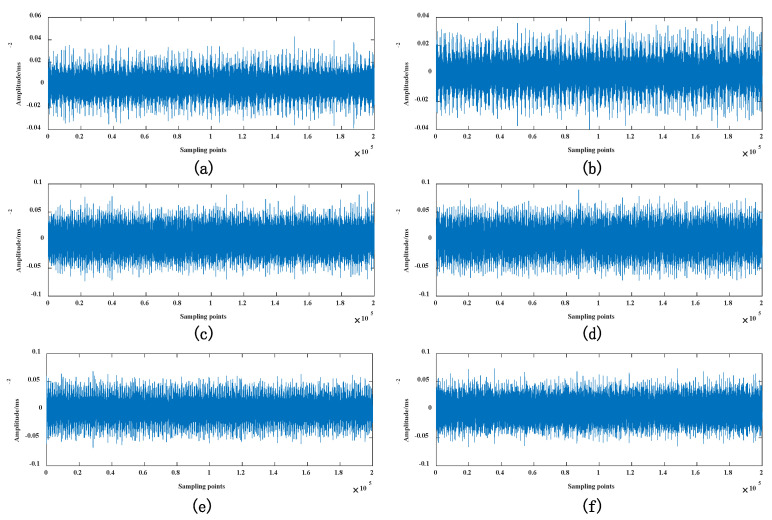
Experimental dataset: (**a**) Valve_800_3mm; (**b**) Valve_800_7mm; (**c**) Valve_1200_3mm; (**d**) Valve_1200_4mm; (**e**) Valve_1200_5mm; and (**f**) Valve_1200_7mm.

**Figure 4 sensors-22-03884-f004:**
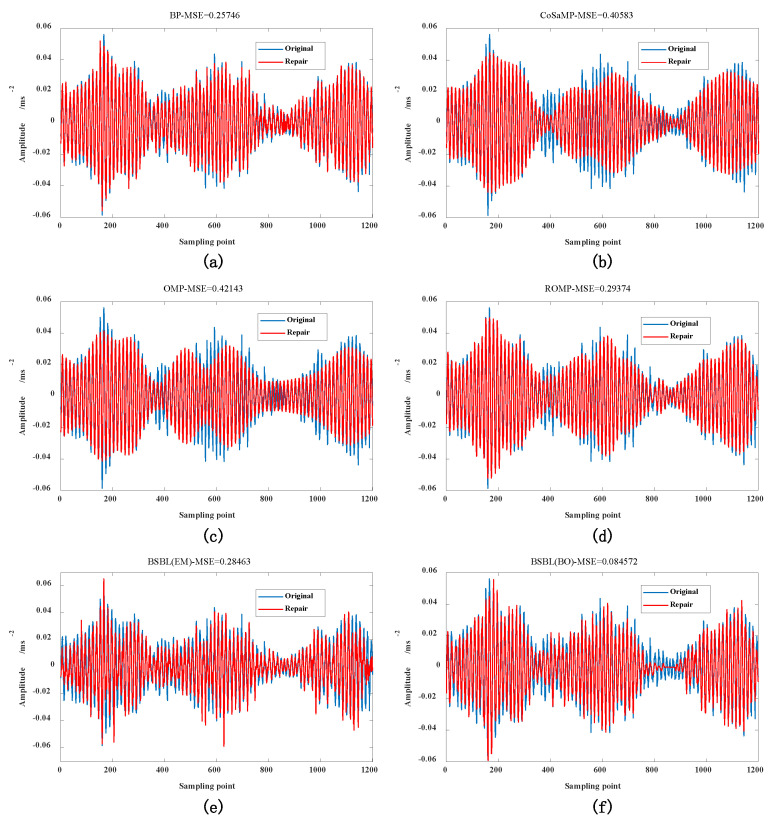
Valve_1200_7mm dataset: (**a**) BP algorithm (MSE = 0.25746); (**b**) CoSaMP algorithm (MSE = 0.40583); (**c**) OMP algorithm (MSE = 0.42143); (**d**) ROMP algorithm (MSE = 0.29374); (**e**) BSBL(EM) algorithm (MSE = 0.28463); and (**f**) BSBL(BO) algorithm (MSE = 0.084572).

**Figure 5 sensors-22-03884-f005:**
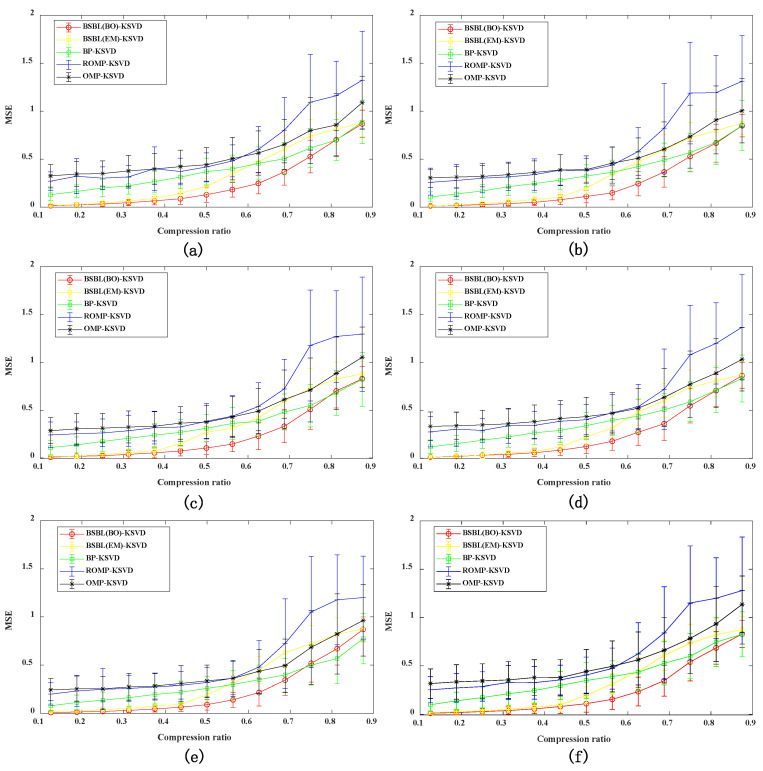
Comparative analysis of MSE 95% confidence intervals of six datasets: (**a**) Valve_800_3mm; (**b**) Valve_800_7mm; (**c**) Valve_1200_3mm; (**d**) Valve_1200_4mm; (**e**) Valve_1200_5mm; and (**f**) Valve_1200_7mm.

**Figure 6 sensors-22-03884-f006:**
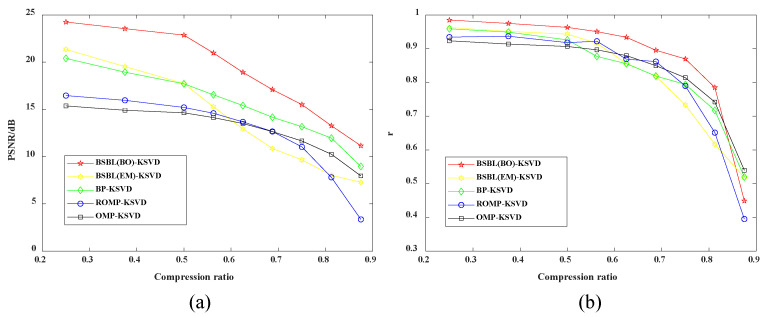
Comparative analysis of different compression and reconstruction methods: (**a**) peak signal-to-noise ratio evaluation and (**b**) Pearson correlation coefficient.

**Figure 7 sensors-22-03884-f007:**
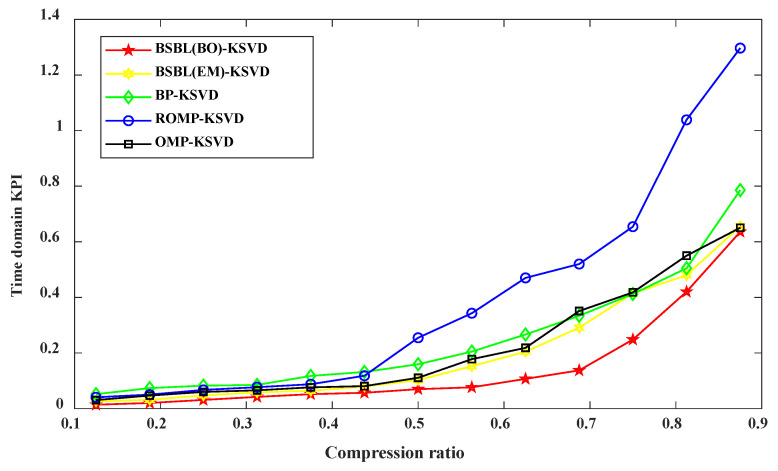
Time-domain comprehensive evaluation index of reconstructed signal under different compression ratios.

**Figure 8 sensors-22-03884-f008:**
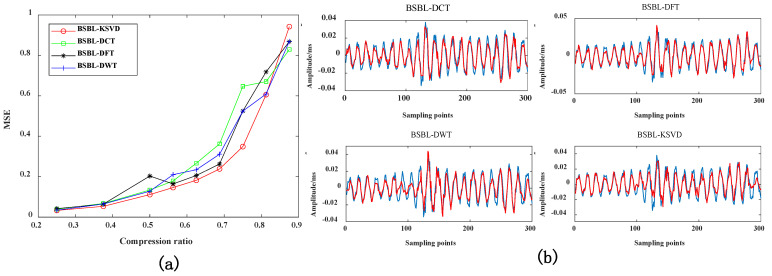
(**a**) Comparison of MSE indicators of different sparse dictionaries; (**b**) when CR = 0.5, the reconstruction signal comparison of different dictionaries.

**Figure 9 sensors-22-03884-f009:**
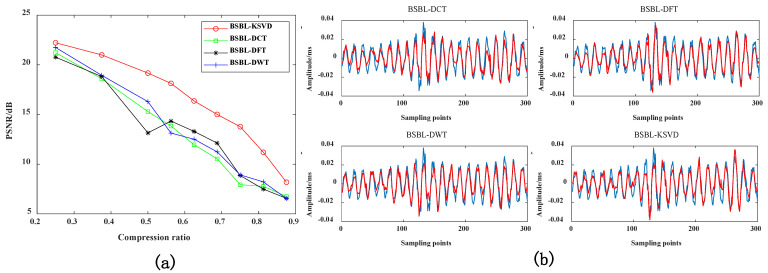
(**a**) Comparison of PSNR indicators for different sparse dictionaries; (**b**) when CR = 0.7, the reconstruction signal comparison of different dictionaries.

**Figure 10 sensors-22-03884-f010:**
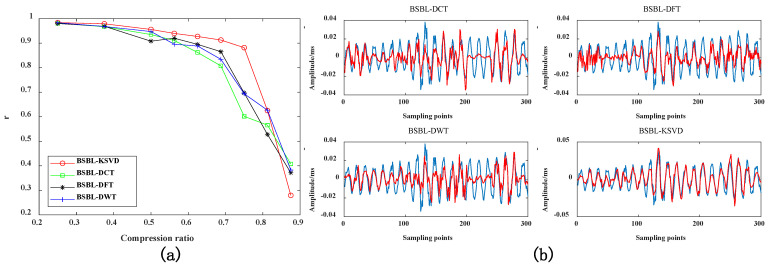
(**a**) Comparison of r indicators of different sparse dictionaries; (**b**) when CR = 0.6, the reconstruction signal comparison of different dictionaries.

**Figure 11 sensors-22-03884-f011:**
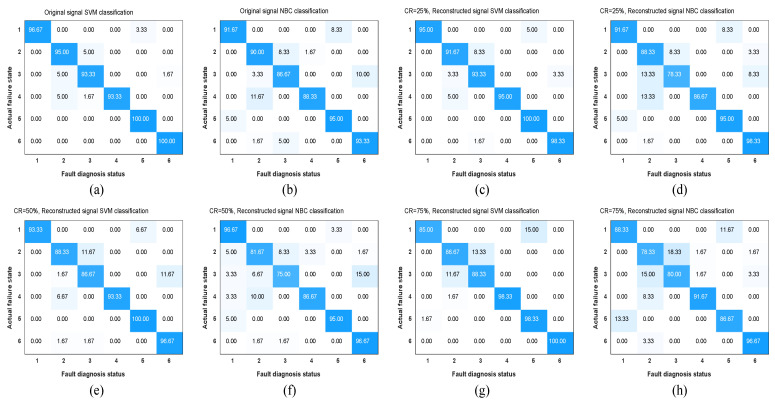
Comparative analysis of fault classification under different compression ratios: (**a**) Original signal SVM classification; (**b**) original signal NBC classification; (**c**) CR = 0.25, SVM classification; (**d**) CR = 0.25, NBC classification; (**e**) CR = 0.5, SVM classification; (**f**) CR = 0.5, NBC classification; (**g**) CR = 0.75, SVM classification; and (**h**) CR = 0.75, NBC classification.

**Table 1 sensors-22-03884-t001:** Experimental Dataset of Valve Clearance in Different Working Conditions.

No.	Dataset	State	Rotating Speed	Inlet Valve Clearance
1	Valve_800_3mm	Normal Status	800	0.3 mm
2	Valve_800_7mm	Fault 1	800	0.7 mm
3	Valve_1200_3mm	Fault 2	1200	0.3 mm
4	Valve_1200_4mm	Fault 3	1200	0.4 mm
5	Valve_1200_5mm	Fault 4	1200	0.5 mm
6	Valve_1200_7mm	Fault 5	1200	0.7 mm

**Table 2 sensors-22-03884-t002:** Comparative analysis of fault classification under different compression ratios.

State	Original Signal		CR = 0.25		CR = 0.5		CR = 0.75	
	SVM	NBC	SVM	NBC	SVM	NBC	SVM	NBC
Normal Status	96.67%	91.67%	95.00%	91.67%	93.33%	96.67%	85.00%	88.33%
Fault 1	95.00%	90.00%	91.67%	88.33%	88.33%	81.67%	86.67%	78.33%
Fault 2	93.33%	86.67%	93.33%	78.33%	86.67%	75.00%	88.33%	80.00%
Fault 3	93.33%	88.33%	95.00%	86.67%	93.33%	86.67%	98.33%	91.67%
Fault 4	100.0%	95.00%	100.0%	95.00%	100.0%	95.00%	98.33%	86.67%
Fault 5	100.0%	93.33%	98.33%	98.33%	96.67%	96.67%	100.0%	96.67%
**Total Accuracy**	**96.39%**	**90.83%**	**95.56%**	**89.72%**	**93.06%**	**88.61%**	**92.78%**	**86.95%**

**Table 3 sensors-22-03884-t003:** When CR = 0.25, the comparative analysis of fault classification of different compression and reconstruction methods.

State	BSBL(BO)-KSVD	BSBL(EM)-KSVD	BP-KSVD	ROMP-KSVD	OMP-KSVD
Normal Status	95.00%	91.33%	83.33%	86.67%	85.33%
Fault 1	91.67%	85.00%	95.00%	86.67%	90.00%
Fault 2	93.33%	83.33%	81.67%	83.33%	82.67%
Fault 3	95.00%	93.33%	88.33%	81.67%	80.67%
Fault 4	100.0%	95.67%	85.00%	95.00%	90.33%
Fault 5	98.33%	100.0%	95.00%	91.33%	92.00%
**Total Accuracy**	**95.56%**	**91.44%**	**88.06%**	**87.45%**	**86.83%**

**Table 4 sensors-22-03884-t004:** When CR = 0.5, the comparative analysis of fault classification of different compression and reconstruction methods.

State	BSBL(BO)-KSVD	BSBL(EM)-KSVD	BP-KSVD	ROMP-KSVD	OMP-KSVD
Normal Status	93.33%	87.67%	78.33%	80.67%	83.67%
Fault 1	88.33%	83.33%	80.00%	82.33%	72.00%
Fault 2	86.67%	85.00%	75.00%	70.33%	72.33%
Fault 3	93.33%	88.33%	81.67%	76.00%	68.00%
Fault 4	100.0%	95.00%	92.33%	87.67%	82.33%
Fault 5	96.67%	90.67%	88.33%	88.00%	85.67%
**Total Accuracy**	**93.06%**	**88.33%**	**82.61%**	**80.83%**	**77.33%**

**Table 5 sensors-22-03884-t005:** When CR = 0.75, the comparative analysis of fault classification of different compression and reconstruction methods.

State	BSBL(BO)-KSVD	BSBL(EM)-KSVD	BP-KSVD	ROMP-KSVD	OMP-KSVD
Normal Status	85.00%	81.00%	72.67%	76.67%	65.00%
Fault 1	86.67%	77.33%	75.00%	65.00%	74.67%
Fault 2	88.33%	77.33%	71.33%	73.67%	67.33%
Fault 3	98.33%	89.00%	83.33%	67.33%	67.33%
Fault 4	98.33%	82.67%	75.67%	70.00%	72.00%
Fault 5	100.0%	85.33%	80.00%	75.67%	63.67%
**Total Accuracy**	**92.78%**	**82.11%**	**76.33%**	**71.39%**	**68.33%**

## Data Availability

Data will be provided upon request.

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
