# Peer review of "Compression Reconstruction and Fault Diagnosis of Diesel Engine Vibration Signal Based on Optimizing Block Sparse Bayesian Learning"

_sensors, 2022, doi:10.3390/s22103884_

Round 1

Reviewer 1 Report

This paper presents a data-driven approach for fault diagnosis and compression reconstruction of a diesel engine vibration signal. The approach is based on block sparse Bayesian learning bound optimization and k-single value decomposition. The method is experimentally validated. The paper may have some merit; however, the writing is poor, which makes the paper very hard to understand. 

* Avoid the use of acronyms in the title. 
* BSBL-KSVD is not defined in the abstract, the abstract must be self-contained.
* Ensure all acronyms are defined in the first use, e.g., BP, OMP, ROMP, EEG, etc. Check it thoroughly.
* What is the motivation for combining BDBL and KSVD?
* Please write clearly and concisely the problem statement as well as the objective of the paper and its contribution.
* There is no mention in the introduction of how the paper connects with fault diagnosis. Include definitions and literature review about fault diagnosis, e.g., A review of convex approaches for control, observation and safety of linear parameter varying and Takagi-Sugeno systems, Processes; A survey of linear parameter-varying control applications validated by experiments or high-fidelity simulations, IEEE Trans. Control Syst. Tech.
* Line 106. "Sampling is performed at twice the frequency of the analog signal"... it should say that sampling is performed at least twice.
* There are a lot of repetitions, and the presentation is poor. For instance, in eq 1, x is defined before and after the equation. 
* The presentation is messy. For instance, in line 156: "In the Formula..." which formula? Moreover, all the paper contains the word formula, replace it with equation.
* Section 2.4. Put your developments in the form of an Algorithm for a better understanding. 
* Improve the quality of the plots in fig 3. 
* Table 1... what does Normal mean? Does it refer to Fault-free? 
* Tabla 1... Columns "Fault location" and "Data sources" do not change. Therefore they can be safely deleted, and the information can be included in the table title.  
* Figure 4... The word repair is wrong. It should be reconstructed.
* Figure 5 is also hard to read. 
* Include a discussion of the percentage of failures that were not correctly classified.
* The paper must be proofread, there are a lot of typos and language issues that must be corrected. 
* The paper has a lot of punctuation errors as well as punctuation along the manuscript. There is also an abuse of upper capital letters.

In summary, the paper requires considerable improvement, and it is very difficult to understand. The writing standard is unacceptable. 

Author Response

Thank you

Reviewer 2 Report

The manuscript "Compression Reconstruction and Fault Diagnosis of Diesel Engine Vibration Signal Based on BSBL-KSVD" focuses on fault diagnosis signal methods based on optimization algorithms such as BSBL-KSVD, BSBL-DCT, BSBL-DFT, BSBL-DWT and compares it's on classic methods.
The abstract was written very poorly. The Authors should use the IMRAD structure in this paper block, too. 
The Introduction section presented a proper background study and it is clear what are the development trends in the field and why current research is required, but the forms of links [5-8] are unacceptable. Please write something different about each cited paper.
The Methodology section is missing. The authors declare some kind of methodology in the second section. But there is no cited research supporting given equations (1) - (5). Is the flowchart shown in Fig.1 developed by the Authors?
The third section described standard indexes and seems adequate to me.
The fourth section focused on experimental results. The structure scheme of the test bench is missing. Please, describe the laboratory equipment. Methods of sampling and processing of results were presented correctness of the research.
The Conclusion section is well-organized and well-writing. 

Author Response

Thank you

Reviewer 3 Report

In this paper, in order to avoid the bandwidth limitation of big data during wireless transmission, a method based on BSBL-KSVD is applied to compress and reconstruct the vibration signal of diesel engine. However, there are still some detail issues need to be elaborated by the authors.

Below are several suggestions and comments:

  1. In line 285-287 “And preset 6 kinds of failure modes of intake valve clearance under different working conditions, the detailed parameters of the data set are shown in Table 1”. However, as shown in Table 1, fault state of Valve_800_3mm is normal.
  2. In Section 4, as authors claim, when the compression ratio is set to 0.5, better data compression and reconstruction effects can be obtained by BSBL-KSVD. However, there are still 100,000 compressed data points under this ratio, is this compressed enough for wireless transmission?
  3. In Section 4.2, why is DCT method uniformly used to generate the sparse dictionary matrix rather than K-SVD method? Would it be more reasonable and logical to use K-SVD method uniformly?
  4. As shown in Figure 5, the MSE of BSBL-KSVD increases with compression ratio. Why do authors claim that “it is proved that when the compression ratio CR=0.5, the compression effect of the BSBL-KSVD algorithm proposed in this paper is optimal, which is more suitable for data compression."
  5. In line 346 "the compression effect of the BSBL-KSVD algorithm proposed in this paper is optimal". But there is no BSBL-KSVD algorithm in Figure 6. Besides, the title of Figure 6 does not correspond with the figure.
  6. Pearson correlation coefficient and peak signal-to-noise ratio are also important indexes to evaluate the effect of data compression and reconstruction. Comparison of BSBL-BO algorithm with other compression and reconstruction algorithms under these two indexes should be supplemented in Section 4.2.
  7. In Section 5, fault diagnosis results based on other data compression and reconstruction methods should be supplemented to further verify the effectiveness of the BSBL-KSVD.
  8. In line 417-418 “Among them, 22 characteristic parameters are extracted for each fault state to form a 22×2160 characteristic matrix.” Please explain how to obtain the 22 characteristic parameters.
  9. What does "select sensor 1-4#" in line 419 refer to?
  10. The meanings of some acronyms used in manuscript are not clarified, such as BP, CoSaMP, OMP, SSAE, etc.
  11. The language is understandable, but there are errors that a native English writer (I am assuming that authors are not), would help to fix easily. I would recommend such a language check.

For example, in line 8 “Using wi-fi communication technology, Remote real-time dynamic monitoring of diesel”, the “Remote” should be “remote”.

  1. The symbols in this paper are not uniform. For example, the symbols in section 2 do not correspond to the symbols in Figure 1.

Author Response

Thank you

Round 2

Reviewer 1 Report

The paper has improved. However, the text still needs to be improved in terms of language, typos, grammar, and punctuation. There are also undefined references as in line 203 "In the equation, Gamma represents...", What equation are you referring to? at which gamma? it should be Y, not Gamma. This type of undefined reference is repeated in the whole document. 

The connection with fault diagnostic must be further emphasized. Include the definition of a fault and more literature analysis is required. What does "The failure analysis of the control system of the diesel engine is carried out under poor working conditions" mean? please extend.

Include the motivation for combining BSBL and KSVD. 

Reviewer 3 Report

The questions proposed have been answered,this paper can be accepted.

Author Response

Thank you